# Towards Embedded Computation with Building Materials

**DOI:** 10.3390/ma14071724

**Published:** 2021-03-31

**Authors:** Dawid Przyczyna, Maciej Suchecki, Andrew Adamatzky, Konrad Szaciłowski

**Affiliations:** 1Academic Centre for Materials and Nanotechnology, AGH University of Science and Technology, Mickiewicza 30, 30-059 Krakow, Poland; suchecki@agh.edu.pl; 2Faculty of Physics and Applied Computer Science, AGH University of Science and Technology, Mickiewicza 30, 30-059 Krakow, Poland; 3Department of Computer Science and Creative Technologies, Unconventional Computing Lab, University of the West of England, Bristol BS16 1QY, UK; andrew.adamatzky@uwe.ac.uk

**Keywords:** concrete, memristors, chaos, reservoir computing, signal classification

## Abstract

We present results showing the capability of concrete-based information processing substrate in the signal classification task in accordance with *in materio* computing paradigm. As the Reservoir Computing is a suitable model for describing embedded *in materio* computation, we propose that this type of presented basic construction unit can be used as a source for “reservoir of states” necessary for simple tuning of the readout layer. We present an electrical characterization of the set of samples with different additive concentrations followed by a dynamical analysis of selected specimens showing fingerprints of memfractive properties. As part of dynamic analysis, several fractal dimensions and entropy parameters for the output signal were analyzed to explore the richness of the reservoir configuration space. In addition, to investigate the chaotic nature and self-affinity of the signal, Lyapunov exponents and Detrended Fluctuation Analysis exponents were calculated. Moreover, on the basis of obtained parameters, classification of the signal waveform shapes can be performed in scenarios explicitly tuned for a given device terminal.

## 1. Introduction

The upcoming era of the Internet of Things may require an energy-efficient pre-processing of signal in order to relieve microprocessors, especially in the case of relatively simple and routine processes. This part of computation can be delegated to solid state computational platforms based on materials, which are not associated with computation, e.g., various construction materials and other parts of the infrastructure. Here, we present a new approach towards unconventional in-materio computation using one of the most ubiquitous materials—Concrete. Small concrete blocks, containing admixtures of dopants (at micro and nanoscale) are demonstrated to present significant computational power to differentiate various waveforms utilizing the principles of reservoir computing. Complex dynamic response of concrete (doped and undoped) can be utilized for computation, and conversely—The dynamic response to known waveforms can be also used for detection of concrete defects, which, in far-fetched vision, can be used for self-monitoring of concrete structures and prevention of fatal accidents.

In developed countries, technology begins to encompass more and more aspects of life. Approximately 87% of humanity has access to electricity, according to the International Energy Agency [1]. In turn, less than half of the population has continuous access to the Internet [2]. Both these percentages increase every year, indicating progressing technological advancement of the human race. Nowadays, a technology that surrounds people with devices connected to the Internet—So-called Internet-of-things (IoT)—is beginning to gain increasing recognition [3]. It can take the form of “smart home”, with connected home appliances, heating, lighting, and wearables of inhabitants with a smartphone or smart speakers in an attempt to increase comfort and security of human life (e.g., in the form of “elder care”). Other applications of IoT include healthcare, transportation, manufacturing, agriculture, or the military. The global market for IoT was valued at 164 mld $ in 2018 [4] and it is predicted that the global market of “smart homes” can reach 58 mld $ in 2020 [5]. At the same time, broadband access to the Internet as well as processing and storage of huge amount of data is extremely important. Fast information processing and storage, however, is an extremely energy-demanding technology. Therefore, at least some of the data processing should be delegated into substrates other than silicon, operating much slower, but at the same time consuming less energy. Alternatively, the waste heat produced during computing can be utilized for heating purposes in colder seasons. This may help to reduce the carbon footprint of computing, which nowadays accounts for 3.2% of the total anthropogenic carbon dioxide emissions [6].

Combination of ideas of *in materio* computing [7,8,9,10,11] and smart houses [12,13,14,15] immediately leads to the concept of computational concrete—Smart material combining construction and information processing features. If successful, such material should render each building an energy-efficient supercomputing device. What if walls would not only support the roof, but at the same time perform advanced, decentralized, and distributed computation? Each building block would sense itself and the environment, monitor safety of the construction, environmental pollution, and interact with humans in an intelligent way. This far-fetched vision has been already proposed and supported with some preliminary experimental and theoretical investigations [16]; however, long-term changes in concrete-based materials, which may lead to significant changes in electric properties of this material, should be taken into account in the case of out-of-lab applications. The selection of concrete as a computational medium seems shocking at the first glance. On the other hand, various unorthodox substrates have been already reported to perform advanced computation, including liquid marbles [17,18], slime molds [19,20], mycelia and fungi [21,22], algae [23], and photochromic solutions [24,25]. In principle, any physical system of sufficiently complex, structure, dynamics, and responsiveness to external stimuli can be utilized for information processing [26,27]. Because of the above, the choice of concrete as a ubiquitous computational medium seems reasonable. Furthermore, concrete is easily and readily prepared and fabricated in all sorts of shapes and structural systems. Its great simplicity lies in the fact that its constituents are ubiquitous and are readily available almost anywhere in the world. As a result of its ubiquity, functionality, and flexibility, it has become by far the most popular and widely used construction material in the world. It is particularly suitable for nano- or micro-modifications due to its peculiar internal structure. The ingredients can be selected, proportioned, and engineered to produce a concrete of specific strength and durability or other multifunctional properties, so it is “fit for purpose” for the job for which it is intended [28]. It can be produced in the form of precast products or as ready-mixed concrete, which is delivered in the familiar rotating concrete lorry. Currently, ingredients are optimized to make concrete strong, light-weight, low-thermally conductive, and durable when exposed to the environment. However new investigations are focused on concrete with embedded sensing [29,30,31,32,33].

Current IoT technology includes a broad set of topics such as sensors, embedded systems and machine learning (ML). ML methods can be used to improve the performance and security of intelligent infrastructure through the prediction of inhabitants’ actions based on their daily behavior [34,35]. This is realized by advanced network systems and software implementation of ML, whereas the building structure acts only as a skeleton to ensure its durability and insulation. Through the use of efficient ML methods, such as Reservoir Computing (RC), it becomes possible to develop intelligent infrastructure based on the building blocks capable of embedded, distributed information processing [16].

RC paradigm can be regarded as an extension of artificial neural networks (ANN) encompassing in its framework various physical substrates and processes [36,37,38]. Its main strength is the so-called “reservoir of states” possessing rich configuration state space of internal dynamics and performing nonlinear transformation of input signals. Thanks to its operation, simplification of the training process of ANN can be achieved, as probing of a reservoir at the readout layer is the only part of the system that needs tuning [39,40]. Probing different features of the reservoir can enable the implementation of pattern recognition, assuming that the given configuration state space is diverse enough [41].

It has been shown by Wlaźlak et.al. that a pure hardware RC system based on a single memristive nonlinear node operating in the delayed feedback loop can be used in the simple classification of signal amplitudes [42,43]. A more complex RC setup based on memristor array (supporting reservoir of states) with ANN software readout was studied as an image recognition system [44] and similar systems were considered for waveform recognition [45]. Therefore appropriate doping, which can induce memristive properties in concrete-based materials is desired. In our recent work, we have suggested the possibility of implementing RC concepts based on hybrid construction material—A “computing concrete” based infrastructure, that could potentially work as a massively interconnected parallel processor [16]. This assumption was drawn on the basis of rich and nonlinear responses of the device to the electrical stimulation. It was theorized that highly nonlinear response arose due to many different pathways for charge carriers and superimposition of capacitive behavior of the device with internal ionic movement. Buildings based on this type of embedded hardware could then possess multisensory properties and support forms of information processing.

This work is aimed at a demonstration that concrete, the most ubiquitous construction material can be used as a computational medium. While semiconducting materials, carbon nanomaterials and various polymers definitely offer better performance [46,47,48], concrete is the most ubiquitous construction material; therefore, despite very limited computational performance, it may be explored and ultimately utilized for some computational tasks. The inherent randomness of concrete seems to be a severe drawback, but materials with a certain degree of irreproducibility can be also studied as a computational medium [49]. Due to limited memristive properties and rather poor internal electrical dynamics, computational tasks in doped concrete can be performed only under a heterotic approach—*in materio* computation must be accompanied by advanced signal analysis using a conventional, software-based approach. Nevertheless, concrete proves to be a useful computational medium capable of basic signal classification.

## 2. Materials and Methods

The base material used in this experiment was ready-to-use concrete mix procured from Leroy Merlin and steel shavings supplied by POCh (Gliwice, Poland). Antimony sulfoiodide nanowires (SbSI) were synthesized in the following procedure. The reactants weighed and added in a ratio of 1 g Sb, 0.265 g S, and 1 g I_2_. All reactants were mixed in a 100 mL flask using 50 mL isopropanol as a solvent. The whole was placed in the ultrasonic bath previously heated to 50 °C for 6 h. The resulting product was isolated by centrifugation at 5000 rpm and washed three times with isopropanol and 2-fold with water after that product was left to dry.

The reference sample consisting of only concrete, as well as modified samples additionally containing 1%, 5%, and 10% of either SbSI, steel shavings, or half and half mixture by weight of both, were created using the following steps. In the bottom of a plastic container, holes 1cm apart were made, creating a 3 × 3 grid. Those holes served as an insertion point for silver wires that would go through the bulk of the material. After preparing the mold, the material was poured in. The whole was firmly shaken to remove pockets of air and allow content to settle within the container. Water was poured until all concreate was sufficiently saturated. Excessive water was drained through entry points of silver wires. The whole was repeatedly shaken to remove any air bubbles that might have appeared. The samples were left to settle and dry at ambient temperature for a week (Figure 1). Samples have been stored and measured in an air-conditioned room with constant temperature (22 °C) and humidity (25%) in order to prevent excessive drying or accumulation of moisture.

Voltammetric and spectroscopic measurements were performed on SP-300 potentiostat (BioLogic, Seyssinet-Pariset, France). Cyclic voltammetry was measured in a −5 V/5 V potential window with a scan rate of 100 mV/s. Electrochemical impedance spectroscopy (EIS) was measured in the 7 MHz–100 mHz frequency window, with 50 mV AC perturbation.

To perform signal mixing in the computing concrete system, two separate arbitrary signals from a dual-channel arbitrary waveform generator (TG5012, Aim-TTi, Huntingdon, Cambridgeshire, UK) were applied via the WA301 waveform amplifier (Aim-TTi, Huntingdon, Cambridgeshire, UK) and impedance matching baluns (1VP-C, Top-View Tek, Shenzhen, Guangdong, China) to two chosen terminals of the sample as indicated in Figure 1. One channel was tuned to 300 Hz with a sinusoidal wave shape, whereas the second channel was tuned to 290 Hz, 280 Hz, and 275 Hz with three different wave shapes for each of these frequencies (sinusoidal, triangular, and square). Application of symmetrical AC signals is crucial, as any DC component of potential higher than that of water electrolysis (approximately 1.23 V) may lead to irreversible changes in the material, even lower DC components may result in electrode corrosion. In that scenario, the sinusoidal signal could be perceived as a base probing signal to classify the second signal of unknown shape in a classification task. Both signals were 10Vpp in amplitude. Processed waveforms were recorded on a digital oscilloscope (DSO-X2014A, Agilent Technologies, Santa Clara, CA, USA). Examples of recorded time series are shown in Appendix A (Appendix A).

Signals recorded at OUT1 terminal were of higher quality, less scattered and were used for further processing. Only in one case (Petrosian fractal dimension), the OUT2 signals were used along with OUT1 ones. Signal processing and analysis were performed in Python (Python Software Foundation, Beaverton, OR, USA). Nolitsa module was used for the time delay (Delayed Mutual Information method) and embedding dimension (False Nearest Neighbors and Average False Neighbors) estimation. By using Nolds (Python module for nonlinear dynamics study) Correlation Dimension, maximum Lyapunov exponent and Detrended fluctuation analysis (DFA) scores were calculated. Further study of dynamical parameters (Petrosian and Katz fractal dimensions, as well as sample and approximate entropy) was performed using the EntroPy Python module for a one-dimensional time series analysis. All analysis was carried out for normalized time series.

## 3. Results

Initially, all obtained samples have been characterized with cyclic voltammetry within ±5 V window. All samples have shown moderate conductivity and currents up to 2.5 mA have been recorded for samples doped with both semiconducting nanowires and metal shavings (Figure 2). It was found that undoped concrete, as well as concrete with low content of any dopant, shows a predominant capacitive hysteresis loop (characteristic for ferroelectric materials) [50,51,52,53] superimposed on Ohmic current. This behavior should be expected for mixed oxide materials [54,55]. The strongest features characteristic for ferroelectric materials has been observed in the case of 10% of SbSI admixture, which is fully consistent with pronounced ferroelectric properties of this material [56,57,58], but this non-ideal capacitive behavior was observed in the majority of cases, the complex character of I/E curves may be interpreted in terms of mixed ferroelectric/antiferroelectric character of studied samples [59]. In light of the complex chemical and phase structure of samples, this may be fully justified. 

Detailed analysis of these phenomena is, however, out of scope of this study. In just few cases memristive behavior was observed, with the most pronounced resistive switching in the case of 10% SM sample (Figure 2). Therefore, this combination of both dopants was selected for further investigations and for the reservoir computing experiments.

Capacitive properties of selected samples were further addressed using impedance spectroscopy. The junction capacitances are low, which can be seen as a decrease of impedance at high frequency region. This effect is less pronounced for doped materials. Moreover, it was found that the Ohmic component increases with increasing concentration of the dopant (Figure 3a). Furthermore, undoped concrete show relatively high phase shift angle at low frequencies (Figure 3b), which can be associated with a Warburg impedance related to a slow diffusion process within a ceramic matrix. Increasing concertation of dopant reduces this contribution because other transport mechanisms start to dominate (Figure 3c).

Based on registered signals (Appendix A), further information processing and analysis were performed based on several methods mentioned vide supra. Due to the lack of control over the spatial arrangement of 3d semiconductor/metallic grains suspended in a cement matrix, geometric change of the place from which we read the signals also changes to some extent the calculated parameters. For this reason, the signal readout layer must be properly calibrated to enable signal classification.

### 3.1. Estimation of Time Delay and Embedding Dimension Parameters

At first, Augmented Dickey–Fuller (ADF) test was calculated to check data stationarity. Results show that for a sample size T = 500 the critical values were not exceeded in any case, the highest *p*-value was obtained for sin/square pair (no more than 1.25%), which means that the null hypothesis can be rejected (that the data possess “unit root” —the presence of stochastic trend) and the time series are in fact stationary [60]. 

According to the Taken’s theorem (which was also shown independently by Packard et al. [61]), single time series can be used to reconstruct so called “delay-coordinate map” based on chosen displacement (time delay) [62,63,64]. Reconstructed attractors possess the same mathematical properties (e.g., Lyapunov exponents, fractal dimensions of the attractor or eigenvalues of a fixed point) as the original manifolds of a given dynamical system (usually obtained on the basis of a set of ordinary differential equations). It comes down to the proper selection of a set of the adjacent coordinates with an equal time offset between them. Classical methods of determining the time delay measure the independence of subsequent points in the phase space. Basically, for infinite, noise-free time series, the selection of time delay can be chosen almost arbitrary [62], but for experimental data, it is good practice to determine its appropriate value. The time delay for the unfoldment of the attractor was estimated using the Delayed Mutual Information (DMI) [65] and Autocorrelation methods [64]. By applying information theory (for which Shannon provided mathematical formalism [66]) to strange attractors, we can quantify the degree of “surprise” new message provides—in the case of attractors these messages are in the form of values given attractor will take during measurement. The DMI method is based on the quantitative approach to uncertainty about time delayed coordinates given the measure of a chosen coordinate. The first minima of calculated functional of joint probability distribution indicates the suitable τ value (Figure 4a). In turn, the first zero of the autocorrelation function gives proper time delay. The autocorrelation method yields the most suitable delay value of three, whereas the DMI method, which is more reliable, yields τ = 4. To inspect the validity of the calculated τ value, several delay times were used to reconstruct attractors in the phase space for a randomly selected data set (Appendix A). It can be observed, that τ = 4 is optimal for the unfoldment of the attractors. It is good practice to choose the smallest time delay required, to avoid phenomena called the irrelevance and redundancy [64,67]. Irrelevance occurs when the reconstructed attractor folds over on itself thus making it more complicated than the original manifold, whereas redundancy means the concentration of attractor shape on the diagonal set. The plot of delayed mutual information versus time delay (Figure 4a) clearly indicates significant chaotic character of all recorded time series with a contribution of a stochastic component. These curves present oscillatory character (fingerprint of chaotic character) and a steep slope as small τ values (stochasticity fingerprint) [68]. 

Based on a calculated time delay, time-delay embedded trajectories have been plotted (Appendix A) [68]. On both sets of trajectories, highly complex system dynamics can be observed. Frequency ratio of applied stimulation influences irregularities in observed traces, which is represented in beats present in the waveforms (Appendix A) and recurring decimal in these frequency ratios. For 300 Hz/290 Hz, recurrence of decimal place is observed for 28 digits, for 300 Hz/280 Hz for seven digits and for 300 Hz/275 Hz frequency ratio for two digits. The attractors are more regular for the cases where there is a smaller number of periodic digits, as well as for a smaller period of observed beats in the registered waveforms. Moreover, with the progressive deviation from the shape of the basic sinusoidal signal, more and more irregular trajectories can be observed (which may be associated with a greater number of harmonic components of the triangular and square wave signals).

In order to analyze the nonlinear dynamics of the recorded time series, the appropriate embedding dimension was determined using the False Nearest Neighbor (FNN) method proposed by Kennel et. al. [69] (Figure 4b) and the Average False Neighbors (AFN) method proposed by Cao [70] (Figure 4c). The FNN method tests whether neighboring points of a specific trajectory in a given embedding dimension are actually neighbors due to the system dynamics or whether they are next to each other only because of the insufficient dimensionality of the phase space. By examining how the number of neighbors changes as a function of dimension, one can determine the appropriate embedding dimension for further analysis. To check the percentage of false neighbors relative to real neighbors, three criterions are used—the first criterion increases the embedding dimension and tests the ratio of Euclidean distance between pairs of points compared to the distance between points with previous embedding dimension value, the second criterion compare relation between reconstructed attractor in higher dimensions and its original size, whereas the third criterion uses both previous tests. Both criteria are compared to a heuristically chosen threshold, values of which are suggested in the original work of Kennel et al. The second condition tries to eliminate the situation where the limited amount of data and the noise present in them causes that the points that are not next to each other are treated as neighbors. To overcome possible problems with choosing proper threshold values in FNN test, Cao proposed his modified FNN method, called AFN or Cao’s test. The main difference is that instead of calculating relative distance ratios separately, a mean value of all of these distances is analyzed between a subsequent increase of embedding dimension (E1(d) in Figure 4c). Cao further defines another testing criterion (E2(d) in Figure 4c), where ratio of mean distances between subsequent embedding dimensions is calculated for the time delayed one dimensional time series and not for reconstructed vectors as in E1(d) criterion. Previously estimated time delay from DMI and autocorrelation methods was used to form time delayed vectors needed in FNN and AFN methods.

FNN results show that the number of false neighbors for all test criteria drops to 0% starting from the embedding dimension of 4 (Figure 4b). This outcome is consistent with the results obtained by the AFN method, where both criteria—E1 and E2—reach saturation starting from the same embedding dimension as in the one indicated in the FNN test (Figure 4c). For this reason, further analysis of nonlinear dynamics was made using the embedding dimension of 4. For a practical reason, however, the trajectories are depicted for embedding dimension of 3 (Appendix A). These figures can be considered as 3D projections of 4D trajectories obtained by the removal of the fourth coordinate.

The complex character of the recorded time series was further characterized with nonlinear dynamics methods (largest Lyapunov exponent), self-similarity methods (detrended fluctuation analysis, fractal dimensions), and disorder-based methods: dynamic (sample entropy) and structural (permutation entropy) entropy-based methods [71].

### 3.2. Analysis of Non-Linear Dynamics

Lyapunov exponents are one of the main indicators of chaos in the study of data possessing non-linear properties [72,73,74]. It probes the rate of divergence of concomitant trajectories in phase space. The exponential rate of divergence of two chaotic trajectories can be described as follows [75] Equation (1):(1)$$\Delta \left(t\right)\sim \Delta_{0}e^{\lambda t}$$
where *λ* is the Lyapunov exponent, and Δ_0_ is the initial separation vector. Due to differences in initial conditions based on a given separation vector, one can obtain a spectrum of Lyapunov exponents. It is common to refer to the largest one as the Maximum Lyapunov exponent (MLE) that is used to probe the predictability and stability of the given data sample. To characterize trajectory instability, MLE can be defined as follows Equation (2):(2)$$\lambda =\underset{t\to \infty }{\lim}\underset{\Delta_{0}\to 0}{\lim}\frac{1}{t}\ln\frac{\Delta \left(t\right)}{\Delta_{0}}$$

Positive MLE strongly indicates the chaotic nature of system dynamics, especially the sensitivity to the initial conditions, which is known as the “Butterfly effect” [76]. Calculated MLE shows, that for seven cases, an un-doped sample presents chaotic behavior (positive MLE) in registered waveforms. In contrast, the doped sample exhibits chaotic behavior in five cases overall (Figure 5a). This is generally consistent with the delayed mutual information dependence on a time delay (Figure 4a), which also indicates chaotic features of the recorded time series. An overall increase of MLE can be observed for sine waveforms, whereas an overall decrease in MLE is present for the square wave shapes.

Detrended Fluctuation Analysis (DFA) is a method for determining the statistical self-affinity of a signal [77,78]. Self-affinity can be regarded as a property of a fractal time series [79]. Using this parameter, one can easily distinguish whether the stimulated sample was doped or not (Figure 5b). Results indicate correlated (α > 0.5) and anti-correlated (α < 0.5) character of the registered time series for undoped and doped samples respectively. Both scaling factors α lie between 0 and 1, indicating the stationary character of the time series (in accordance with ADF results, vide supra). Furthermore, those results indicate the presence of memory in registered time series, [77] which is consistent with the presence of measured memristive traces. The anti-correlated character of time series registered from doped samples may originate from the possibility of flipping resistive state, observed in CV measurements (Figure 2).

Another parameter used in the study of chaotic and dynamical systems is the correlation dimension (*ν*) [80]. It is used to probe dimensionality of the space occupied by a set of random points and is often referred to as a type of fractal dimension. For time series of points described as Equation (3):(3)$${\left\{\vec{X\left(i\right)}\right\}}_{i=1}\equiv {\left\{\vec{X\left(t+i\tau \right)}\right\}}_{i=1}$$
where *τ* is arbitrary, but fixed time increment. The correlation integral is defined as Equation (4):(4)$$C\left(r\right)=\underset{N\to \infty }{\lim}\frac{1}{N^{2}}\sum_{i,j=1}^{N}\Theta \left(r-\left|\vec{X\left(i\right)}-\vec{X\left(j\right)}\right|\right)$$
where Θ(X) is a Heaviside step function. For small number r, correlation integral behaves according to a power law, Equation (5):(5)$$C\left(r\right)\sim r^{v}$$
where *v* is interpreted as a fractal dimension [80,81]. As can be seen in Figure 5c, the change of correlation dimension is strongly correlated with the shape of mixed signals. The correlation dimension is only shifting downwards for the mixing of sin/sin signals, only upwards for triangle signals, whereas for square (280–290 Hz) it shifts downwards and for 275 Hz *v* shifts upwards. Based solely on this fact, classification of signal shape can be performed (overall *v* decrease: sine, overall *v* increase: triangle, mixed *v* trend: square).

### 3.3. Classification of a Waveform on the Basis of the Decision Tree Method

As was already mentioned, changing the readout terminal influences obtained dynamical parameters. With this change, analysis of obtained parameters allows for different classification scenario. Analysis of Sample Entropy [82,83] and fractal dimensions (vide infra) gives an alternative approach towards signal classification. Sample entropy is a technique used for probing regularity/complexity (unpredictability of fluctuations) of time-series signals. It possesses desirable characteristics in the form of data length independence and relatively trouble-free implementation. It is defined as a negative natural logarithm of conditional probability between distances of two sets of points taken from template vector which acts as a representation of a given data sample.

For time series, Equation (6):(6)$$N=\left\{x_{1},x_{2},x_{3},\ldots ,x_{N},\right\}$$
the template vector takes a form of Equation (7):(7)$$X_{m}\left(i\right)=\left\{x_{i},x_{i+2},x_{x+2},\ldots ,x_{i+m-1},\right\}$$
where m is embedding dimension. Based on this, sample entropy can be described as Equation (8):(8)$$S_{S}=-\ln\frac{A}{B}$$
where *A* and *B* are numbers of template vector pairs having distance ($d\left[X_{m+1}\left(i\right),X_{m+1}\left(j\right)\right]$ and $d\left[X_{m}\left(i\right),X_{m}\left(j\right)\right]$, for *A* and *B*, respectively) lower than given tolerance r (which is taken as a factor of standard deviation). 

If analyzed data is ordered, then templates for m points are also similar for *m*+1 points, and *A/B* approaches unity [83]. In that case, the negative logarithm will approach 0. Results show, that in most cases (Figure 6a, apart from 300 Hz/290 Hz sin-square), obtained time series are more ordered for doped sample, which may be associated with less noise present in the signal (cf. Appendix A).

Trends observed in sample entropy changes (Figure 6a) do not allow unambiguous classification of waveforms, therefore other criteria must be used in parallel.

Permutation entropy is considered as a natural measure of time series complexity via reconstruction of a phase space of any dynamic system [84,85]. Here it was calculated according to Yan et al. [86] according to the Takens–Maine theorem. The phase space of a time series {x(i), i=1, 2, 3,…, N} can be reconstructed as Equation (9):(9)$$\{\begin{array}{c} X\left(1\right)=\left\{x\left(1\right),x\left(1+\tau \right),\ldots ,x\left(1+\left(m-1\right)\tau \right)\right\}\\ \ldots \\ \begin{array}{c} X\left(i\right)=\left\{x\left(i\right),x\left(i+\tau \right),\ldots ,x\left(i+\left(m-1\right)\tau \right)\right\}\\ \begin{array}{c} \ldots \\ X\left(N-\left(m-1\right)\tau \right)=\left\{x\left(N-\left(m-1\right)\tau \right),x\left(N-\left(m-2\right)\tau \right),\ldots ,x\left(N\right)\right\} \end{array} \end{array} \end{array}$$
where *m* is the embedded dimension and *τ* is the time delay. Then, the m number of real values contained in each X(i) can be arranged in an increasing order as Equation (10):(10)$$\left\{x\left(i+\left(j_{1}-1\right)\tau \right)\leq x\left(i+\left(j_{2}-1\right)\tau \right)\leq \ldots \leq x\left(i+\left(j_{m}-1\right)\tau \right)\right\}$$

If there exist two or more elements in X(i) that have the same value, e.g., $x\left(i+\left(j_{1}-1\right)\tau \right)=x\left(i+\left(j_{2}-1\right)\tau \right)\;$, their original positions can be sorted in such a way that for $j_{1}\leq j_{2}$ the relation $x\left(i+\left(j_{1}-1\right)\tau \right)\leq x\left(i+\left(j_{2}-1\right)\tau \right)$ will be obtained. Hence, any vector X(i) can be mapped onto a group of symbols, Equation (11):(11)$$S\left(l\right)=\left(j_{1},j_{2},\ldots ,j_{m}\right)$$
where *l* = 1, 2, …, *k* ≤ *m*!. S(l) is one of the *m*! symbol permutations, which is mapped onto the *m* number symbols $\left(j_{1},j_{2},\ldots ,j_{m}\right)$ in *m*-dimensional embedding space. If *P*_1_, *P*_2_, …, *P_k_* are used to denote the probability distribution of each symbol sequences, respectively, and the condition described by Equation (12):(12)$$\sum_{l=1}^{k}P_{l}=1$$
is fulfilled, the permutation entropy of a time series {x(i), i=1, 2, 3,…, N} can be defined as a Shannon entropy for the *k* symbol sequence, Equation (13):(13)$$S_{p}\left(m\right)=-\sum_{l}^{k}P_{l}\ln P_{l}{}_{}$$

As the maximum value of $S_{p}\left(m\right)$ for a uniform probability distribution is equal to lnm!, it is usually given as a normalized value, Equation (14):(14)$$S_{p}\left(m\right)=\frac{-\sum_{l}^{k}P_{l}\ln P_{l}{}_{l}}{\ln m!}$$

The values of permutation entropy serve as a measure of time series randomness. Smaller values indicate less chaotic behavior whereas values approaching the unity indicate highly chaotic behavior and thus unpredictability of time series. These data suggest that time series recorded for sine and triangle waveforms are significantly more ordered (i.e., less chaotic) than those for square waves, which may be utilized as a classification tool (Figure 6b).

Analysis of Petrosian [87] and Katz fractal dimensions [88] allows a different approach for signal classification. Katz fractal dimension (*D_K_*) calculates the fractal dimension of data directly from the waveforms without the need for their abstract representation. It is defined as Equation (15):(15)$$D_{K}=\frac{\log_{10}n}{\log_{10}\frac{d}{L}+\log_{10}n}$$
where *d* is calculated fractal dimension, *n = L/a* (n is used for normalization of distances —L is the total sum of lengths of the successive points, and a is averaged distance between successive points) and *d* is the maximum distance between the first point and any other point within the data set. *D_K_* is known to overestimate probed fractal dimension, hence large differences in obtained *D_K_* and correlation dimension scores [89]. Figure 5c shows calculated Katz fractal dimensions of attractors for various waveform and frequency combinations.

Another approach in probing fractal dimension of time series was suggested by Petrosian [87]. Petrosian fractal dimension (*D_P_*) is calculated for binarized time series. It is defined as follows in Equation (16):(16)$$D_{P}=\frac{\log_{10}N}{\log_{10}N+\log_{10}\frac{N}{N+0.4N_{\delta }}}$$
where *N* is the length of the time series, and *N_δ_* is the number of sign changes in the signal derivative. It can be observed in Figure 7, that Petrosian fractal dimensions increase in the series sine<triangle<square for both output signals and both materials. There is, however, a significant change in the undoped/doped difference, as indicated by black arrows in Figure 7.

## 4. Discussion

Each individual dynamics index cannot serve as a reliable classification index for waveform discrimination. Therefore, observed trends, taken collectively, constituted a set of criteria that can be used for waveform discrimination on the basis of the signal dynamics in pristine and heavily doped concrete blocks. These criteria, along with the dynamic analysis presented above, can be regarded as a readout layer of the reservoir computing system. 

Based on different trends of change of the given parameter between doped and undoped samples (Table 1), one can classify signal shapes in a decision tree manner. A decision tree could be constructed as follows:√If calculated permutation entropy decreases and Katz fractal dimension is of mixed trends, then the signal is of the triangle wave shape.√If calculated Petrosian fractal dimension is increasing, then the signal is of sinusoidal shape, if its decreasing (and was increasing in the previous step), then it is of square shape.

As already mentioned, changing the readout terminal influences obtained dynamical parameters (vide supra). With this change, the analysis of obtained parameters allows for different classification scenarios. Based solely on the Petrosian fractal dimension of registered time series but analyzed from two different device terminals (OUT1 and OUT2, Figure 7) another classification variant of a decision tree manner can be obtained.

A decision tree based on trends summarized in Table 2 could be constructed as follows:If the calculated Petrosian fractal dimension (OUT1) is increasing, then the signal is of sinusoidal shape, if its decreasing (and was increasing in the previous step), then it is of square or triangular shape.If calculated Petrosian fractal dimension (OUT2) is increasing, then the signal is of triangular shape, if its decreasing (and was decreasing in the previous step), then it is of square shape.

Along with various trends (changes in various dynamic parameters upon transition from pristine to doped concrete) another classification system, based on the whole collection of time series can be also derived (Figure 8). Three selected criteria provide the best classification of waveforms, and also provide means for the classification of concrete material. Interestingly, detrended fluctuation analysis yields exponent α, which can differentiate between doped and undoped concrete, but does not provide means for signal classification. Time series recorded for pristine concrete are much higher (α > 0.50) than for doped concrete (α < 0.25). This indicates a statistically higher correlation of time series for pristine material and anticorrelation for doped one. This may be associated with quite different dielectric responses of both materials. Sample entropy (*S_s_*) is not a useful classification criterion, both due to the same trend over all samples (vide supra) and due to very scattered values (Figure 8). Petrosian fractal dimension for sine and triangular waveforms are significantly lower than for square signals, therefore it may serve as a crude criterion for detection of square wave signals. Finally, the permutation entropy provides a weak classification tool for all waveform shapes: sine waves yield the lowest values, triangular waves the intermediate ones, whereas square waves the highest values of *S_p_*. This criterion should be considered as a fuzzy one, as the boundary between sine and triangular waves is not well defined.

Despite the requirement of a complex numerical processing of the data in order to extract the classification parameters the results presented here clearly indicate that computation with appropriately prepared concrete blocks is possible. Surprisingly, concrete—one of the most ubiquitous construction materials—shows complex chaotic dynamics when stimulated with acoustic frequency electrical signals. Moreover, these dynamics can serve as a classification tool. The selection of a wider range of frequencies and waveforms should lead to more complex classification patterns. It seems, that concrete itself presents internal electrical dynamics so complex, that in principle it should be capable of much more complex computational tasks in real-time. Recently reported speech recognition in coupled nano vortex oscillators [90] is based on a system of comparable dynamics (however shifter to radio frequency range). Therefore, any classification of acoustic signals required their mapping into radio frequencies. The system presented here performs complex classification tasks directly on amplified signals.

The device presented in this paper (Figure 9) can be regarded as a heterotic reservoir computing system. The heteroticity originates from combination of *in materio* reservoir processing of input signals followed by software algorithms for post-processing. In a far-fetched vision, an alternative, *in materio*-based readout should be considered, but the required complexity of signal processing seems to exceed the state of the art of *in materio* reservoir computers. The observed features indicate, that the small concrete blocks with silver wire electrodes show a set of features sufficient for reservoir computing. The fading memory feature is represented by capacitive and memristive character of the device, whereas internal dynamics are provided by the drive signal. It shows the echo state property, as the output at the selected point reflects features on inner electrical dynamics. The dynamic response of the system is complex enough to provide sufficient separability (in the sense of Stone–Weierstrass theorem [91]) of the input data [92]. It also presents some generalization features, as the observed output space (trends of several criteria) is much smaller than the (infinite) input space of various signals. Due to the specific task and material properties of pristine and doped concrete the output layer, especially the post-processing part, is relatively complex. It should be noted, however, that this was the requirement for a relatively hard task for memristive reservoir computing systems and that the memristive properties of deliberately chosen materials were very poor. Despite this, the classification task is successfully performed. Future upgrade of this system may involve fuzzy logic inference engine (or multinary logic), as the output trends are not crisp values, and therefore the fuzzy descriptors may be more adequate. Interestingly, multinary and fuzzy logic may be also implemented in related materials [93,94,95].

The computational performance of concrete would depend, however, not only on the doping state but many other factors: humidity, temperature, and the age of samples. The dependence of performance on environmental factors can be used for various sensing purposes, while aging processes may irreversibly change the computing performance of concrete artifacts. Although this may not be beneficial from an information processing point of view, it is very interesting in itself and may also find applications in the future as a tool for monitoring the status of concrete structures.

Aging of concrete involved both chemical changes (hydrolysis and corrosion of silicate network) as well as self-weight consolidation of the granular skeleton. This leads to changes of hydro-mechanical properties and displacements of the ensembles of functional nanoparticles. If the “computing circuits” (i.e., the assemblies of metallic and/or semiconducting particles) in concrete were fixed the ageing would be detrimental to the computational properties of the concrete. However, the computing circuits prototypes presented in this work are highly distributed (hence can be regarded as amorphous), and therefore deformations or even topological modifications of the ensembles of functional particles should not significantly affect the computing abilities of the concrete materials. The interfacial and external computing elements will be continuously adjusting to the ever-changing morphologies of the concrete. Moreover, a reconfiguration of the embedded computing circuits during concrete ageing might be advantageous because novel computational properties will emerge. This may be taken as a distant analogy to the nervous system, which at different stages of development shows different functionalities: intensive learning abilities at an early age, peaking classification in adulthood, and slowly declining memory in older age. 

Along with the classification of signals, the concrete-based reservoir computing device can be used in a reversed way (cf. Figure 8). One can consider the doping state as the input and the specific responses to electrical stimulation as the output. Due to the fact that the fading memory echo state machines are universal approximators, as analyzed in detail by Ortega et al. [91,96,97], they can operate easily in a reversed way. This reversed operation of reservoir computer may be ultimately used to infer the state of the concrete sample. This would require a set of standard stimulation protocols along with a database of responses corresponding to the different states of concrete samples. Although this has not been addressed experimentally, there are numerous reports on the use of electrical measurements (e.g., impedance spectroscopy) for monitoring of concrete elements [98,99,100]. We can envision, that combination of classical electrical measurements with signal-processing features of concrete can lead to more efficient concrete monitoring techniques and may contribute to increased safety and prevention of fatal accidents. However, this far-fetched vision requires years of further intense studies in the fields of concrete technologies and reservoir computing.

## 5. Conclusions

In this article, the classification of signal shapes was shown based on *in materio* computing concrete hardware system. Samples present a highly non-linear response in regard to data transformation, possess rich configuration state space, and their dynamics (when stimulated with a simple sine wave drive) are represented in the form of four-dimensional trajectories. These features make them a suitable platform for reservoir computing implementation. Depending on used terminals for the readout layer, different classification scenarios can be achieved. Moreover, as it can be seen in the case of the detrended fluctuation exponent, some of the chaos indices can be used for the classification of the doping state of concrete. Whereas this feature has not been explored deeply in this study, in principle it may constitute a new tool for real-time monitoring of concrete structures, detection of structural degradation and prevention of fatal accidents. The presented results can be treated as proof of the concept for the possibility of information processing and classification tasks performed by appropriately doped ubiquitous construction materials. 

It should be noted, however, that the performance of concrete-based computing devices is very limited. Classification of waveforms in concrete can be performed only with the support of time-series processing algorithms operating at classical computers. Therefore, the presented results can be regarded as an approach towards concrete-based heterotic computing. On the other hand, the same numerical algorithms cannot perform signal classification due to fundamental reasons: chaos indices (like Lyapunov exponents, signal entropies and fractal dimensions) are not suited for the classification of smooth periodic signals. The concrete samples themselves also cannot perform this type of classification. Combination of two apparently incompatible approaches results in successful computation. It also illustrates the theoretical concept addressing the balance between the complexity of the information processing systems and the interface. In the studied case the processing system is simplified to the extreme (almost a piece of stone); therefore, the post-processing and readout must be more complex [41].

Further development of the concept can bring the realization of more aspects of a multisensory infrastructure capable of information processing based on its embedded hardware and intelligent computing houses as a far-fetched vision.

## Figures and Tables

**Figure 1 materials-14-01724-f001:**
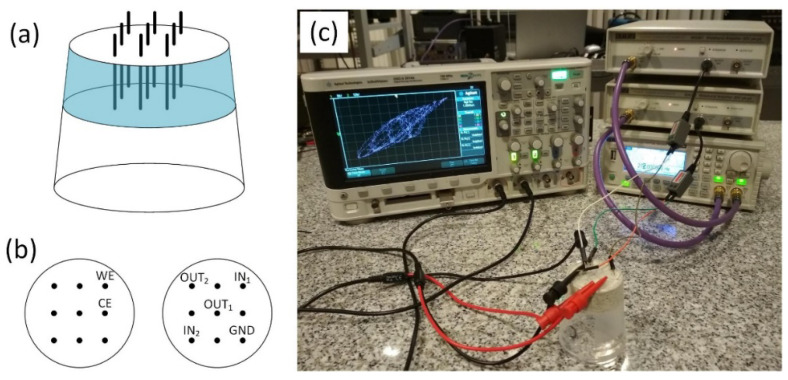
Schematic view of the computing concrete sample (**a**), pinouts for voltammetric (left) and signal processing experiments (**b**) and a real photo of an experimental setup (**c**). WE and CE stands for working and counter electrodes, respectively. IN1 and IN2 are signal input connectors, OUT1 and OUT2 output ones, GND is a common ground.

**Figure 2 materials-14-01724-f002:**
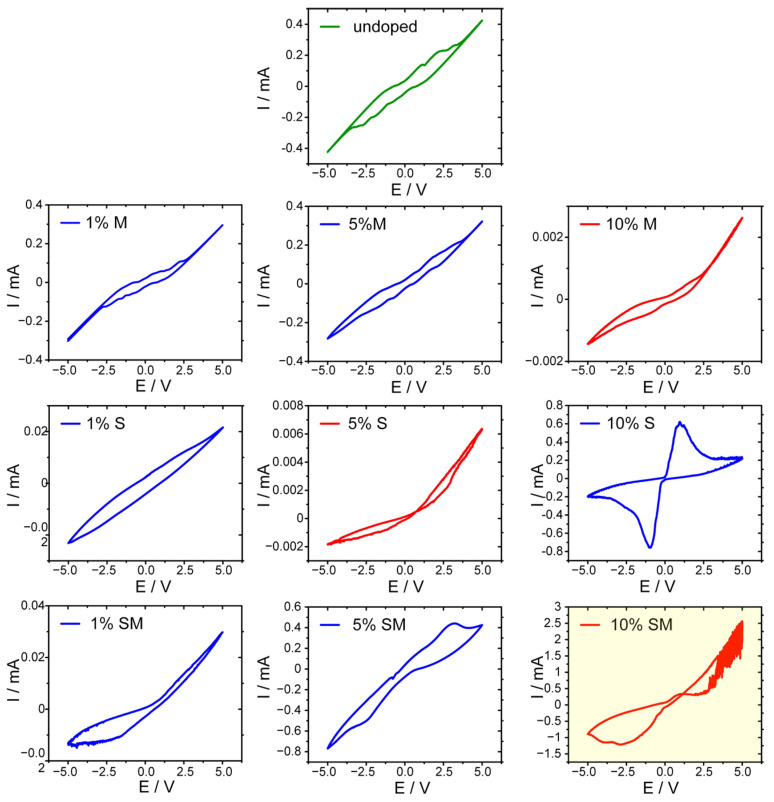
Voltamperometric characteristics of undoped concrete sample (top, dark green) and concrete containing various amounts of dopants: M—Metal shavings, S—Antimony sulfoiodide nanowires, SM—1:1 mixture of both dopants. Some samples show pinched hysteresis loops typical for memristive devices (red) whereas the others are of capacitive character (blue). The most pronounced memristive behavior was observed in the case of 10% SM sample, highlighted in yellow.

**Figure 3 materials-14-01724-f003:**
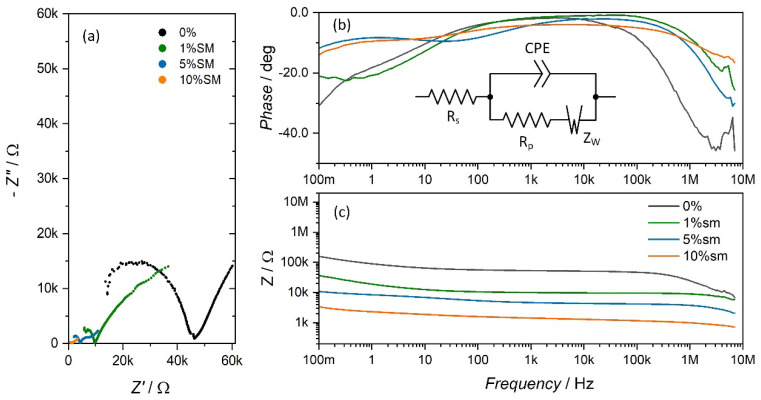
Impedance spectra of undoped and metal+semiconductor doped concrete samples: Nyquist plot (**a**) phase shift angles (**b**) and Bode plots (**c**). A simplified equivalent circuit is also shown. The linear Warburg component at low frequencies is visible only in the case of the undoped sample, whereas increased doping is correlated with a decrease of impedance results as well as with significant curvature of the low-frequency arm in Nyquist plots.

**Figure 4 materials-14-01724-f004:**
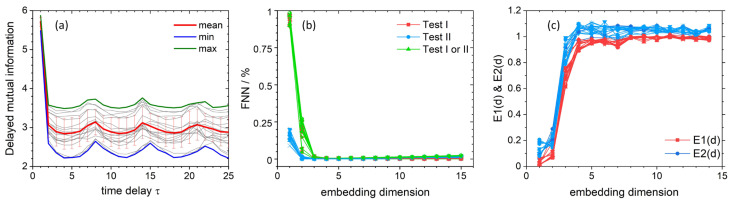
Results for time delay and embedding dimension calculations for all-time series recorded for pristine and doped (10%MS) concrete samples. Graphs present results from Delayed Mutual Information approach (**a**), False Nearest Neighbors test (**b**) and Average False Neighbors method (**c**). Calculated time delay τ = 4 (first minima of DMI, averaged over all data sets), whereas suitable embedding dimension equals four (0% of FNN in all tests and saturation of E1 & E2 in AFN). Descriptions of test criteria can be found in the text.

**Figure 5 materials-14-01724-f005:**
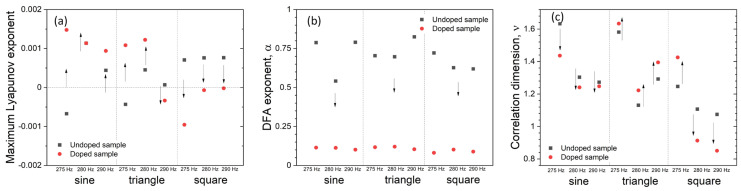
A set of dynamics indices for recorded time series. Maximum Lyapunov exponent for of a time series recorded for different input frequencies/waveforms (**a**). Detrended fluctuation analysis performed for time series recorded for different input frequencies/waveforms (**b**). Correlation dimension of a time series calculated for different input frequencies/waveforms (**c**). Arrows indicate the direction of changes upon doping.

**Figure 6 materials-14-01724-f006:**
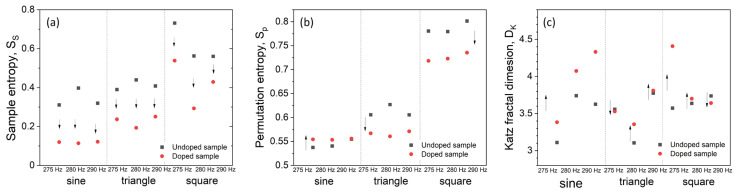
A set of dynamics indices for recorded time series. Sample entropy for different input frequencies/waveforms (**a**). Permutation entropy for different input frequencies/waveforms combinations (**b**). Values of Katz fractal dimension score for doped and un-doped sample different input frequencies/waveforms (**c**). Arrows indicate the direction of changes upon doping.

**Figure 7 materials-14-01724-f007:**
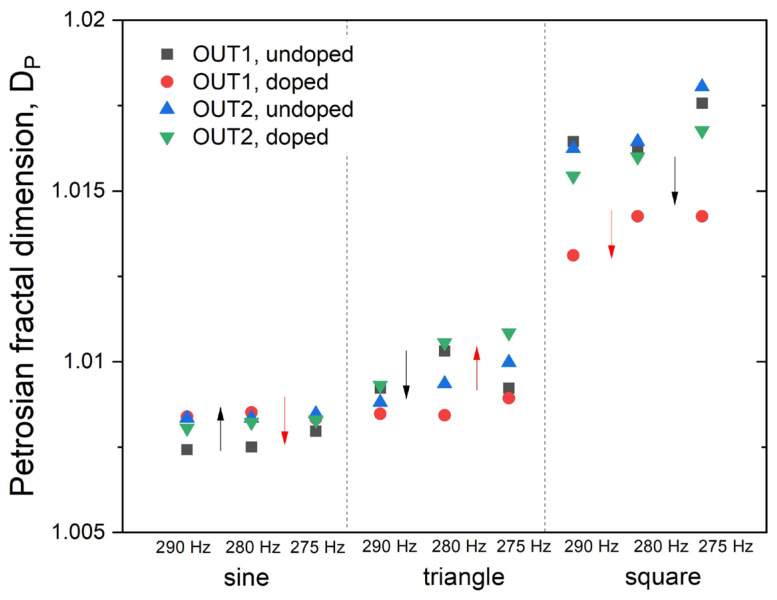
Petrosian fractal dimension for time series collected from OUT1 (**a**) and OUT2 (**b**) device terminals. Black arrows indicate trends for OUT1, whereas red arrows for OUT2 (cf. Figure 1 for terminal markings).

**Figure 8 materials-14-01724-f008:**
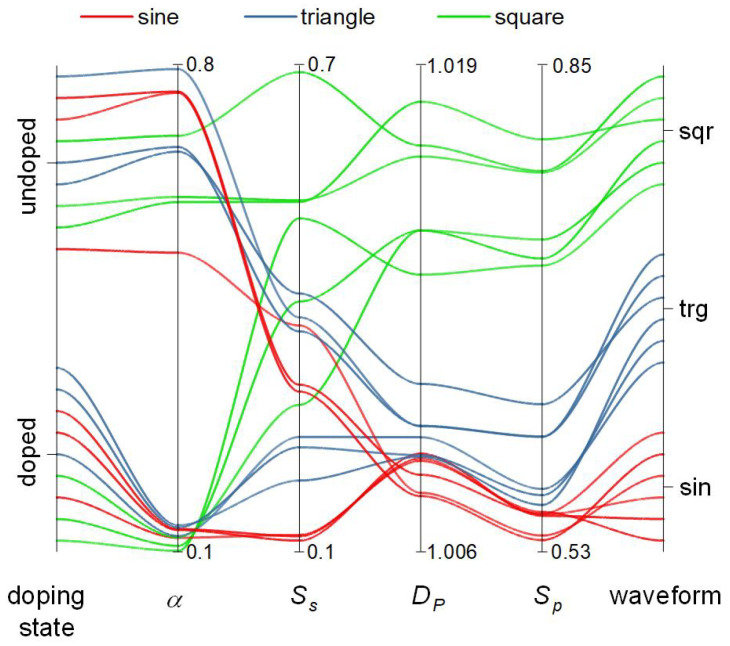
Parallel coordinate plot for all-time series and selected dynamic criteria: detrended fluctuation exponent (*α*), sample entropy (*S_s_*), Petrosian fractal dimension (*D_P_*) and permutation entropy (*S_p_*). Detrended fluctuation exponent can serve as a classification factor for the concrete doping state, whereas permutation entropy significantly classifies time series according to their waveforms.

**Figure 9 materials-14-01724-f009:**
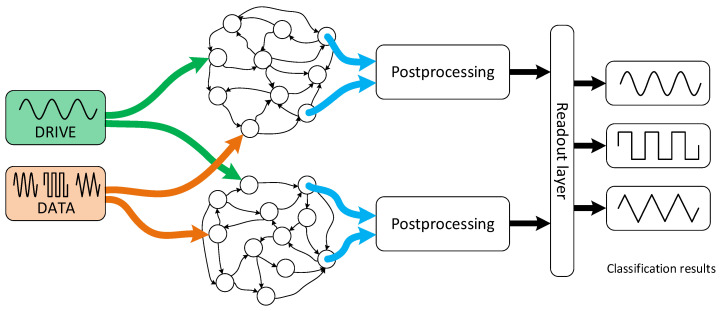
A scheme of a dual concrete-based reservoir computing system used for waveform classification.

**Table 1 materials-14-01724-t001:** A list of trends observed between different shapes of mixed signals for different methods of analysis. Trends are shown for the doped sample in relation to the un-doped one.

Chaos Index	Sine	Triangle	Square
Permutation entropy	increases	decreases	increases
Katz fractal dimension	increases	mixed	increases
Petrosian fractal dimension	increases	decreases	decreases

**Table 2 materials-14-01724-t002:** A list of trends observed between different shapes of mixed signals for different methods of analysis. Trends are shown for the doped sample in relation to the un-doped one.

Chaos Index	Sine	Triangle	Square
Petrosian fractal dimension (OUT1)	Increases	Decreases	Decreases
Petrosian fractal dimension (OUT2)	Decreases	Increases	Decreases

## Data Availability

The data that support the findings of this study are available from the corresponding author upon reasonable request. All time series as well as electrochemical and impedance spectroscopy data are available as asci as well as instrument-specific binary files.

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
