# Peer review of "Towards Embedded Computation with Building Materials"

_materials, 2021, doi:10.3390/ma14071724_

Round 1

Reviewer 1 Report

Summary:

The paper presents an in-materio computing paradigm using doped concrete as substrate for the physical computing. An evolution algorithm controlled by a classical computer is used to configure the physical computer to be able to discriminate between different tone frequencies and waveforms. The output of the physical computer is postprocessed using different statistical techniques that allow to discriminate between the three different input waveforms. Results show that the postprocessing is also able to determine the doping state of concrete. The paper concludes that doped concrete with a mixture of metallic shavings and antimony sulfoiodide nanowires at 5%-5% concentrations has desirable characteristics to be used for in-materio computing, nevertheless due to the poor memristive properties obtained with the proposed mixture a postprocessing using traditional computing was still needed. Future developments of this material could allow to use structural concrete in buildings as physical computers imbedded in constructions.

Broad comments:

The reviewer thanks the authors for the clearly written paper in a very interesting topic.

  1. What is the paragraph between the abstract and the introduction?
  2. The notion of building computing network mentioned in the abstract (line 5) is far from nowadays state of the art and difficult to imagine given the results of the paper. Considering the long-term evolution of concrete (it has been shown that concrete continues to hydrate decades after) the evolutionary training of a given computer would probably become outdated quickly as the material ages.
  3. Using concrete properties like conductivity to monitor building and structure health is a different concept than physical computing and it is not easy to see how the two can be related. At most, degradation of concrete properties due to aging could be viewed as a malfunction cause for a physical computer using this concrete as substrate. How do you imagine the use of physical computing in buildings for this matter?
  4. Line 174: a 10Vpp signal is used as input, did you check if this voltage can cause some irreversible damage to the concrete sample? for instance dehydration of the cement hydrated compounds, pore water hydrolysis, corrosion, sulfation… water hydrolysis is of special concern since the sample has been just air dried for a week meaning that it still contains hygroscopic water and water hydrolysis potential is 1.23V. A possible solution for this could be to dry the sample at 105 degrees and apply a wax seal to avoid humidity variations. No need to repeat the experiments for this publication but the possible limitations and impact on the results should be mentioned.
  5. Lines 548-550: Isn’t the analysis of output fractal dimension a somehow complex computational task that blurs the essence of in-materio computing? Wouldn’t it be possible to directly discriminate the wave signals with those techniques without undergoing in-materio processing? In line 572 the possibility of a materio-readout is considered to exceed the state of the art of nowadays reservoir computers, nevertheless, we can find several examples in the literature using carbon nanotubes and liquid crystal that successfully discriminate signal frequencies fully in-materio, did you mean concrete reservoir computers in particular?
  6. Overall the paper needs to highlight the advantages of using concrete as a physical computing substrate, for this it is needed to specify what are the properties of concrete compared to other common substrates like PMMA or PBMA and possible advantages and disadvantages, the vague idea of decentralized building computer network is not strong enough to justify the use of concrete as a substrate.

Author Response

On behalf of all authors I would like to thank for detailed critical comments, which has helped us to improve the manuscript.

  1. What is the paragraph between the abstract and the introduction?

Due to an editorial error the caption „Introduction” was shifted between the first and second paragraph of the introduction. The error has been corrected.

  1. The notion of building computing network mentioned in the abstract (line 5) is far from nowadays state of the art and difficult to imagine given the results of the paper. Considering the long-term evolution of concrete (it has been shown that concrete continues to hydrate decades after) the evolutionary training of a given computer would probably become outdated quickly as the material ages.

The notion of „building computing” has been removed from the abstract and more detailed discussion of future concepts of application of computing concrete has been added to the revised manuscript. We agree that aging and other changes in the structure of concrete will modify (or even disable) the computational performance of concrete, appropriate comments on consequences have been added. Please note that the manuscript is a proof-of-principle and is meant to pave the way towards unconventional applications of concrete and related materials. A note on the impact of concrete aging has also been added to the discussion.

  1. Using concrete properties like conductivity to monitor building and structure health is a different concept than physical computing and it is not easy to see how the two can be related. At most, degradation of concrete properties due to aging could be viewed as a malfunction cause for a physical computer using this concrete as substrate. How do you imagine the use of physical computing in buildings for this matter?

This issue has been better explained on the basis of Figure 8. In the studied case sensing can be understood as a reversal of the signal classification scheme – various transformations of probing signals may be used for monitoring of the structure/properties of concrete in a similar way as impedance spectroscopy is used.

  1. Line 174: a 10Vpp signal is used as input, did you check if this voltage can cause some irreversible damage to the concrete sample? for instance dehydration of the cement hydrated compounds, pore water hydrolysis, corrosion, sulfation… water hydrolysis is of special concern since the sample has been just air dried for a week meaning that it still contains hygroscopic water and water hydrolysis potential is 1.23V. A possible solution for this could be to dry the sample at 105 degrees and apply a wax seal to avoid humidity variations. No need to repeat the experiments for this publication but the possible limitations and impact on the results should be mentioned.

The reviewer is right – application of electric signals may lead to electrolysis and electrode corrosion. Therefore, in our experiments we have applied symmetrical AC signals of relatively high frequencies to prevent water electrolysis and corrosion. Symmetry of the input signal was preserved by inductive baluns compensating any undesired impedance mismatch, which may result in loss of signals’ symmetry. Samples have been prepared in an air-conditioned laboratory with humidity control to prevent excessive changes in water content in studied samples, details are included in the experimental part.

  1. Lines 548-550: Isn’t the analysis of output fractal dimension a somehow complex computational task that blurs the essence of in-materio computing? Wouldn’t it be possible to directly discriminate the wave signals with those techniques without undergoing in-materio processing? In line 572 the possibility of a materio-readout is considered to exceed the state of the art of nowadays reservoir computers, nevertheless, we can find several examples in the literature using carbon nanotubes and liquid crystal that successfully discriminate signal frequencies fully in-materio, did you mean concrete reservoir computers in particular?

The analysis of fractal dimensions, Lyapunov exponents and various time series entropies has been performer using open source Python libraries, and from numerical point of view complexity of these calculation is not very high. On the other hand, these techniques applied to input data (pure sine, square or triangular waves or their combinations) cannot be used for waveform classification. All smooth periodic signals have (by definition) fractal dimension and correlation exponents of unity and zero entropies and Lyapunov exponents. Therefore, these numerical techniques cannot be used for successful determination of their waveform. In the studied case additional noise and nonlinear transformation of input signals facilitates their classification.

The unclear passage on reservoir computing is re-written. We just wanted to say that in materio implementation of analyses presented in this manuscript (entropies, fractal dimensions, Lyapunov exponents) exceeds the state of the art. We are aware that reservoir computing is a powerful technique and other implementations may perform advanced signal analysis. Concrete as a computing material has numerous limitations, but is interesting as it is one of the most ubiquitous materials.

  1. Overall the paper needs to highlight the advantages of using concrete as a physical computing substrate, for this it is needed to specify what are the properties of concrete compared to other common substrates like PMMA or PBMA and possible advantages and disadvantages, the vague idea of decentralized building computer network is not strong enough to justify the use of concrete as a substrate.

The application of concrete as a computational medium originates from its ubiquitesness. We are aware that there are materials much better suited for signal processing, but concrete has been suggested as a computing medium for primary signal sensing and processing for future and far-fetched applications in smart houses. Furthermore, the presented example is an illustration of the thesis that even a rock can compute. As it has been noticed, computation with a piece of concrete requires powerful interface with complex readout functions. It is an experimental proof of a theoretical investigations by Konkoli et al. who has demonstrated that decreasing complexity of the dynamic system must be accompanied with increased complexity of the readout layer. Detailed analysis of the overall efficiency in the case of heterodic approach towards computation fall out of the scope of current manuscript. Anyway, we would like to thank reviewer for turning out attention to this problem.

Reviewer 2 Report

Review of the article "Towards embedded computation in building materials".
The authors of the article: Dawid Przyczyna, Maciej Suchecki, Andrew Adamatzky, Konrad Szaciłowski

Manuscript ID: materials-1134355

The manuscript contains theoretical and experimental substantiation of new technology embedded computing in building materials. The manuscript contains five sections, supplementary information and a list of references.

Section 1 introduces a new systemic understanding of the concepts of a smart home, the Internet of Things (IoT) and the rationale for the author's concept of computational concrete - smart combination of materials..

Section 2 describes the experimental part of the study in sufficient detail.

Volt-ampere characteristics (Figure 2) and other theoretical and experimental results of Section 3 illustrate the possibility of technical implementation of the author's approach and present the results demonstrating the possibility of processing information based on specific data in the problem of signal classification in accordance with the of material calculations paradigm.

Section 4 discusses in detail a set of criteria that can be used for waveform recognition based on the basis of the signal dynamics in pristine and heavily doped concrete blocks.

In section 5, in logical connection with the previous sections, it is rightfully noted that the  study result presented in the manuscript could become a new tool for monitoring concrete structures in real time, detecting structural degradation and preventing accidents.

Thus, the study is relevant and of scientific and practical interest. The experimental and theoretical results correspond to the physical meaning of the problem; the modeling algorithms are theoretically substantiated. The conclusions are logically consistent with the presented rationales and arguments.

The work is well structured; all parts of the manuscript are logically interconnected. The text is clear and easy to read, arouses interest.

The literature list is quite complete, contains both new (2020) and earlier publications on the research topic.

However, there are minor comments on the work.

  1. The purpose of the work in section 1 is not clearly defined.
  2. Typo in line 95: …the whole system. [33,34] This is done from …

Reviewer's opinion: the article is recommended for publication after a small revision.

Author Response

On behalf of all authors I would like to thank for detailed critical comments, which has helped us to improve the manuscript.

1. The purpose of the work in section 1 is not clearly defined.

Section 1 has been corrected, the last paragraph is focused on the aim of work. The overall discussion of the results and conclusion sections has been also carefully corrected.

  1. 2. Typo in line 95: …the whole system. [33,34] This is done from …

All typos have been checked and corrected.

Round 2

Reviewer 1 Report

Summary

The paper presents an in-materio computing paradigm using doped concrete as substrate for the physical computing. An evolution algorithm controlled by a classical computer is used to configure the physical computer to be able to discriminate between different tone frequencies and waveforms. The output of the physical computer is postprocessed by classical computers using different time-series algorithms that allow to discriminate between the three different input waveforms. Results show that the postprocessing is also able to determine the doping state of concrete given a known input waveform. The paper concludes that doped concrete with a mixture of metallic shavings and antimony sulfoiodide nanowires at 5%-5% concentrations has desirable characteristics to be used for in-materio computing, nevertheless due to the poor memristive properties obtained with the proposed mixture a postprocessing using traditional computing was still needed. The paper presents a proof of concept and performance of concrete computers is still limited compared to other existing in-materio computers, nevertheless the approach is very promising given the simplicity of the used physical substrate.

Broad comments:

The reviewer thanks the authors for the reply and the modifications of the manuscript. The modified version is an improvement in the sense that it clearly establishes the contribution and limitations of the approach. The reviewer thinks that this is a very promising field that will gain attention in the future and the present paper is an important contribution to it.

The main modifications of the revised version are as follows:

Lines 72-74: Concrete evolution challenges are correctly addressed.

Lines 149-160: Ubiquitous characteristic of concrete is highlighted.

Lines 193-195:  Possible corrosion and damage effects by the electric currents are addressed.

Lines 659-693: Limitations of the approach are addressed.

Lines 676-678: "Reconfiguration of the circuits during concrete aging could discover new properties"; this is a very interesting concept and it resembles to the learning process and reconfiguration of human brain during aging.

Lines 713-725: Highlights the strengths of combining classical and physical computers and also the simplicity of the physical subtract used in this study.

Line by line comments:

Line 83 Typo: Because